



# Radar sounding survey over Devon Ice Cap indicates the potential for a diverse hypersaline subglacial hydrological environment

Anja Rutishauser[1*], Donald D. Blankenship[1,2], Duncan A. Young[1], Natalie S. Wolfenbarger[1], Lucas H. Beem[2], Mark L. Skidmore[2], Ashley Dubnick[3], and Alison S. Criscitiello[3]

[1]Institute for Geophysics, University of Texas at Austin, Austin, TX 78758, USA
[2]Department of Earth Sciences, Montana State University, Bozeman, MT 59717, USA
[3]Department of Earth and Atmospheric Sciences, University of Alberta, Edmonton, Alberta, Canada

*Correspondence to*: Anja Rutishauser (rutishauser.anja@gmail.com)

**Abstract.** Prior geophysical surveys provided evidence for a hypersaline subglacial lake complex beneath the center of Devon Ice Cap, Canadian Arctic; however, the full extent and characteristics of the hydrological system remained unknown due to limited data coverage. Here, we present results from a new, targeted aerogeophysical survey that provides evidence (i) supporting the existence of a subglacial lake complex and (ii) for a network of shallow brine/saturated sediments covering ~170 km$^2$. Newly resolved lake shorelines indicate three closely spaced lakes covering a total area of 24.6 km$^2$. These results indicate the presence of a diverse hypersaline subglacial hydrological environment with the potential to support a range of microbial habitats, provide important constraints for future investigations of this compelling scientific target, and highlight its relevance as a terrestrial analog for aqueous systems on other icy worlds.

## 1 Introduction

While numerous presumably freshwater subglacial lakes have been identified beneath the Antarctic (Siegert et al., 2016; Wright and Siegert, 2013) and Greenland ice sheets (Palmer et al., 2013; Howat et al., 2015; Willis et al., 2015; Bowling et al., 2019), a recent study showed evidence for a unique hypersaline subglacial lake complex beneath Devon Ice Cap (DIC), Canadian Arctic (Rutishauser et al., 2018). The features identified as subglacial lakes are situated in two bedrock troughs (T1 and T2) in the cold-based interior of the ice cap (Burgess et al., 2005; Van Wychen et al., 2017; Paterson and Clarke, 1978) where basal ice temperatures are expected to be well below the pressure-melting point of ice (Figure 1). The brine-rich fluid of the lakes is hypothesized to originate from the dissolution of a salt-bearing geological unit, referred to as the Bay Fiord Formation and abbreviated as Ocb (Harrison et al., 2016; Mayr, 1980; Thorsteinsson and Mayr, 1987) that is projected to outcrop at the ice-bed interface in the vicinity of the subglacial lakes (Rutishauser et al., 2018).



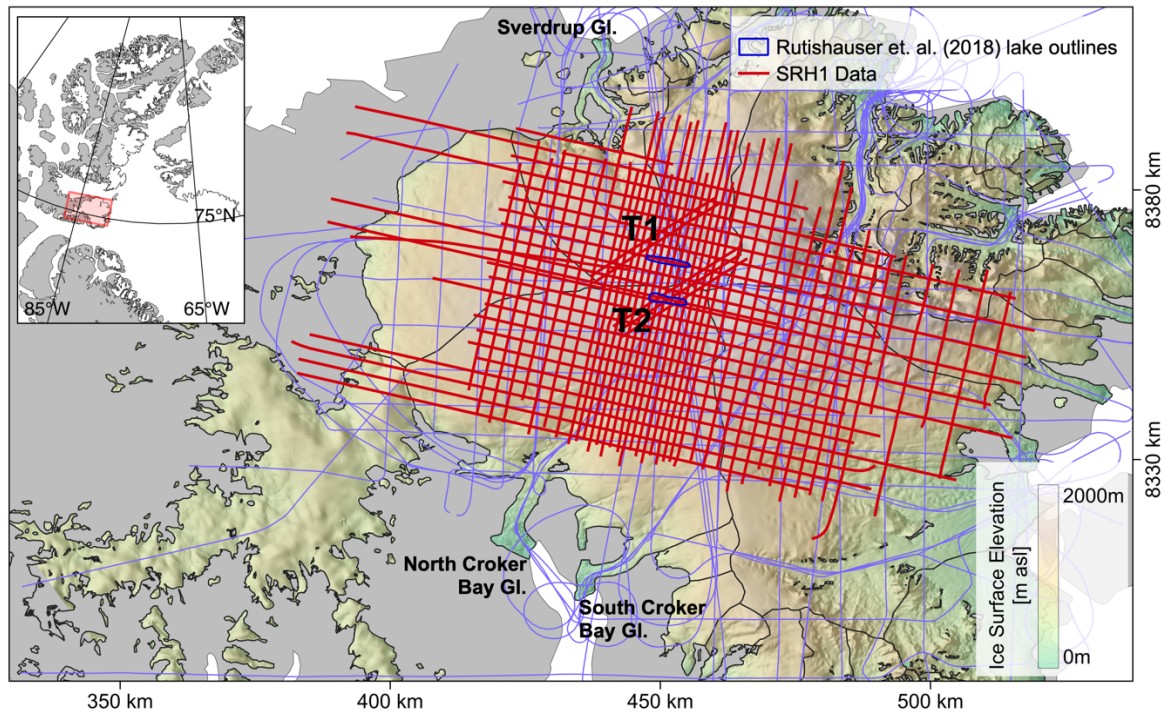

**Figure 1:** Map of Devon Ice Cap overlain with existing radar datasets collected prior to our survey (blue), including data presented in Dowdeswell et al. (2004), Rutishauser et al. (2016, 2018), as well as conducted in Operation Ice Bridge surveys between 2011-2015 (IceBridge MCoRDS L2 Ice Thickness, Version 1, 2021), and the aerogeophysical survey profiles collected in this study (SRH1; red). Blue outlines mark the location of the previously inferred subglacial lakes in bedrock troughs T1 and T2 (Rutishauser et al., 2018) and the thin black lines mark the boundaries of the glacier catchment areas. Background image features the ArcticDEM surface elevation map by the Polar Geospatial Center from DigitalGlobe Inc. imagery (ArcticDEM).

The Devon subglacial lakes were inferred from radar sounding measurements, a tool that has been widely used to characterize subglacial hydrological conditions (Carter et al., 2007, 2009; Young et al., 2016; Palmer et al., 2013; Chu et al., 2018; Schroeder et al., 2015; Bowling et al., 2019). However, the nature and full extent of the lakes and the surrounding hydrological conditions remained unresolved due to the relatively sparse data coverage. This motivated a targeted airborne geophysical survey over DIC where 4415 km of profile lines were collected in dense survey grids (Figure 1). Here, we use the resulting radar sounding measurements to evaluate the two features previously identified as subglacial lakes (Rutishauser et al., 2018), examine their full extents, and characterize the surrounding subglacial hydrological environment. Our results support the previous interpretation for one of the subglacial lakes (Rutishauser et al., 2018) and indicate large areas of wet bed consistent with projected salt-bearing rock outcrops beneath the ice where the other lake was identified (Rutishauser et al., 2018), which we interpret as a brine network rather than a lake. We estimate the areal extent of the lakes and distributed brine network and model potential flow routes of the subglacial brine. Finally, subglacial hydrologic systems, both fresh and saline, have been shown to harbor unique microbial ecosystems (Mikucki and Priscu, 2007; Karl et al., 1999; Skidmore et al., 2005; Christner



et al., 2014; Boetius et al., 2015; Achberger et al., 2017), and have therefore long been considered as terrestrial analogs for icy habitats on other planetary bodies (Cockell et al., 2013; Garcia-Lopez and Cid, 2017). Here, we discuss the microbial habitats that could be hosted in the diverse subglacial environment beneath DIC and the relevance of this system as a terrestrial analog for aqueous systems on other icy worlds.

## 2 Data and methods

Data used in this study were collected during an aerogeophysical campaign over DIC in spring 2018 (dataset referred to as SRH1), utilizing a Basler BT-67 (DC-3T) aircraft operated by Kenn Borek Air Ltd as survey platform. A total of 4415 km along-track data were acquired in a grid survey with line spacing ranging from 1.25 km to 5 km and the densest grid centered over the area of the previously inferred subglacial lakes (Figure 1). Radar Echo Sounding (RES) data were collected using the Multifrequency Airborne Radar-sounder for Full-phase Assessment (MARFA) (Young et al., 2016), a dual-phase center version of the High Capability Airborne Radar Sounder (HiCARS) system operated by the University of Texas Institute for Geophysics (UTIG). The radar is a coherent system with a 60 MHz center frequency and a 15 MHz bandwidth (5 m wavelength in air). Detailed instrument characteristics and processing techniques are described in Peters et al. (2005, 2007). Additional science instrumentation deployed during the SRH1 survey included a Novatel SPAN Inertial Navigation System for precise positioning and orientation, a Riegl laser altimeter, a Cesium vapor magnetometer, and a downward looking Canon DSLR camera. In this study, we present the radar sounding data and use unfocused SAR processed low-gain data to derive basal reflection coefficients, 1-D focused SAR processed data (Peters et al., 2007) to identify the subglacial bedrock topography utilized in the bed digital elevation model (DEM) and basal roughness estimates, and combine 1-D and 2-D focusing to derive the specularity content of the bed echo (Schroeder et al., 2013, 2015). The nominal along-track trace spacing of the dataset is about 22 m with a vertical resolution of about 6 m in ice, and an average pulse-limited footprint diameter at the glacier bed of 274 m.

### 2.1 Bedrock DEM

The DEM of the bed topography previously derived in Rutishauser et al. (2018) compiled RES data collected over DIC by the Scott Polar Research Institute (SPRI) in 2000 (Dowdeswell et al., 2004), HiCARS data collected in 2014 (Rutishauser et al., 2016, 2018) and RES data from Operation Ice Bridge surveys between 2011-2015 (IceBridge MCoRDS L2 Ice Thickness, Version 1, 2021). Here, we update the previous bed DEM (1x1 km grid mesh) using the SRH1 data as an additional dataset and produce a new DEM over a 500x500 m grid mesh.

The bed return from the SRH1 data is identified using a semi-automated picking algorithm, which locates the maximum bed reflection power within manually-defined depth boundaries. Over the steep valley walls of bedrock trough T2, the basal



reflection was discriminated from cross-track clutter using the results from Scanlan et al. (2020). Travel times were then converted to depths using a radar wave velocity in ice of 168.4 m/µs.

Bed elevation data from all surveys over DIC described above were interpolated over a 500 m grid mesh via ordinary kriging to generate a new DEM. To ensure a continuous transition between the bed DEM and the non-glaciated surrounding topography, land elevations from the ArcticDEM, Polar Geospatial Center from DigitalGlobe Inc. imagery (ArcticDEM) outside of the ice cap were included in the bed DEM generation. A total of 47233 crossover points reveal a mean cross-over error of 17.6 m (median of 7.6 m) and a standard deviation cross-over error of 42.5 m. Sources of errors in the radar-derived bed elevation include uncertainties in the ice surface due to a heterogeneous firn affected by melting/refreezing processes (Rutishauser et al., 2016) that are propagated into the bed elevation, cross-track scattering at the surface and bedrock (Scanlan et al., 2020), and uncertainties in the velocity-to-depth conversion. A comparison of the bed DEM to the raw data shows that largest gridding errors appear in deep bedrock troughs connected to the ice cap's outlet glaciers and where the ice is thicker (Fig. S2). These errors are likely caused by uncertainties in the bedrock picks due to cross-track reflections at the subglacial valley walls (Scanlan et al., 2020), a decreased signal-to-noise ratio beneath thicker ice, and a rapidly changing bed topography over short distances not captured in our grid.

## 2.2 Basal reflectivity

Radar-derived measurements of basal reflectivity have been widely used to identify the presence of subglacial water (Jacobel et al., 2009; Peters et al., 2005; Chu et al., 2018; e.g. Carter et al., 2007). The basis for such interpretations is that an ice-water interface has a higher reflectivity than surrounding areas where ice is in direct contact with dry bedrock. However, since bed reflectivity values derived from radar measurements are also affected by the characteristics of the radar system and attenuation processes, a number of corrections are required before basal reflectivity values can be interpreted in terms of subglacial hydrological conditions (Wolovick et al., 2013; Chu et al., 2016; Schroeder et al., 2016; Matsuoka et al., 2010, 2012). Here, we derive the relative basal reflectivity (R) following

$$[R]_{dB} = [P]_{dB} + [B]_{dB} + [G]_{dB} + [L]_{dB} - [S]_{dB} , \qquad (1)$$

where $P$ is the returned bed power, $B$ is the birefringence effects due to variations of the ice crystal fabric (Matsuoka et al., 2003), $G$ is the power loss from geometric spreading of the radar beam, $L$ is the loss from englacial attenuation, and $S$ is the correction for power variations in the radar system, where the notation $[]_{dB}$ refers to the terms expressed in decibels ($[X]_{dB} = 10log_{10}(X)$) (Matsuoka et al., 2012). Here, $S$ is assumed to be constant as no changes were made to the radar instrument settings during the field campaign. Under the assumption of a relatively uniform pattern of crystal fabric orientation over the survey area, we assume that birefringence effects are relatively constant, and thus neglect both terms $S$ and $B$ when analyzing relative basal reflectivity. The power loss from geometric spreading of a specular echo is derived from

$$[G]_{dB} = 2\big[2\big(h + d/\sqrt{\varepsilon}\big)\big]_{dB} , \qquad (2)$$





where $\lambda$ is the radar wavelength in air, $h$ is the aircraft range above the glacier surface, $d$ is the ice thickness, and $\varepsilon = 3.17$ (Evans, 1965) is the dielectric permittivity of ice (e.g. Schroeder et al., 2016). The englacial attenuation loss term $L$ is related to the one-way depth-averaged attenuation rate N via

$$[L]_{dB} = 2Nh . \tag{3}$$

Attenuation rates were derived from a linear fit between the ice thickness and geometrically corrected bed reflection power
(e.g. Gades et al., 2000; Schroeder et al., 2016; Wolovick et al., 2013). We constrain our fit to ice thicknesses between 50-650 m where the relationship appears most linear (Figure 2a), resulting in a one-way attenuation rate of 21.8 dB/km. We estimate an attenuation rate uncertainty of  3.1 dB/km (Figure 2b), using the half width ($N_h$) of the correlation coefficient minimum following Schroeder et al. (2016). The uncertainty of the measured basal reflectivity is estimated from the mean crossover error of the geometrically corrected bed power values along the survey profiles, which is 5.2 dB.


In this study, we do not attempt to derive absolute basal reflection coefficients, but rather compare basal reflectivity in a relative sense. To allow for an easier visual inspection of reflectivity anomalies that could suggest the presence of subglacial water, we normalize the basal reflectivity by subtracting the mean of all measured reflectivity values such that they are distributed around 0 dB (Figure 4f).


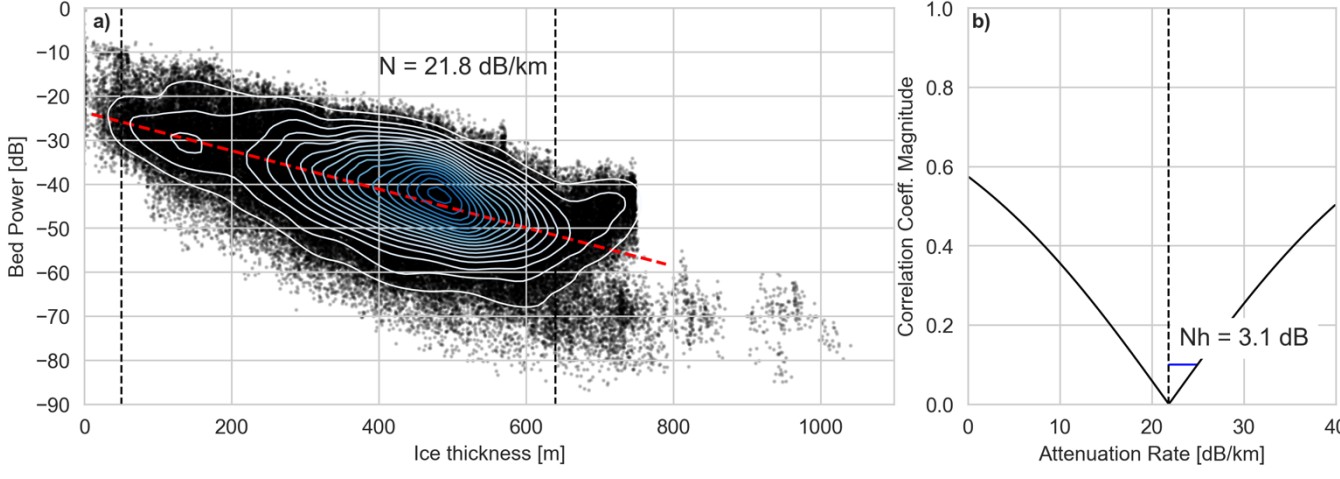

**Figure 2. Derivation of englacial attenuation rates**. a) Correlation and linear fit (red) between the ice thickness and the geometrically corrected bed power of the SRH1 dataset over DIC, along with contour lines indicating the density distribution. b) Attenuation rate versus correlation coefficient magnitude for the SRH1 dataset (Schroeder et al., 2016). We use the half-width of the correlation coefficient minimum
($N_h$) to estimate the attenuation rate uncertainty.

## 2.3 Specularity content

The specularity of the returned bedrock signal is governed by scattering properties of the radar wave, and is sensitive to the interface roughness on a wavelength scale (Shepard and Campbell, 1999; Schroeder et al., 2015). The scattering characteristics



can be derived from the shape (i.e. pulse-peakiness/waveform abruptness) of the reflected waveform (Oswald and Gogineni,
2008, 2012; Jordan et al., 2017; Cooper et al., 2019) or as a function of the along-track angular distribution of the returned
energy (Schroeder et al., 2013; Young et al., 2016; Schroeder et al., 2015), the latter of which we apply here. Ice-water
interfaces that are flat on wavelength scales are expected to produce specular radar reflections (Schroeder et al., 2013, 2015),
a characteristic that has previously been used to identify subglacial water (Oswald and Gogineni, 2012; Greenbaum et al.,
2015; Rutishauser et al., 2018; Carter et al., 2007; Young et al., 2016) and characterize the configuration of subglacial
hydrological systems (Schroeder et al., 2013, 2015). Additionally, the specularity has been used to derive the "fine-scale"
roughness of glacier beds (Jordan et al., 2017; Cooper et al., 2019), which has been related to the subglacial geology (Cooper
et al., 2019).

Here, we derive the specularity by comparing the returned peak energy from two different SAR focusing aperture lengths (1-
D and 2-D focusing corresponding to 0.1 µs and 1 µs range delay, respectively) following Schroeder et al. (2015). To ensure
the interpretation of subglacial water only over significantly large areas, we apply a running-mean filter to the basal reflectivity
and specularity over a 250 m window length along-track, corresponding to just below the average pulse-limited footprint at
the glacier bed of our dataset.

**2.4 Basal roughness**

While the specularity content is a proxy for basal roughness on a wavelength scale, we derive larger-scale basal roughness via
the root-mean square deviation (RMSD) of the bedrock topography along flight lines following Shepard et al. (2001). We
tested different lags (step sizes) between 50 m and 1000 m over 5 km window lengths repeated at every bedrock observation,
for which the resulting subglacial roughness pattern does not change significantly. To show the distribution of the RMSD-
derived subglacial roughness in this study, we choose a lag of 500 m.

**2.5 Subglacial hydraulic head and water flow paths**

We calculate the hydraulic head over a 500 m mesh grid following Shreve (1972) and described in Wolovick et al. (2013),
using ice surface elevation derived from the ArcticDEM (ArcticDEM), the newly generated bed DEM, and a subglacial brine
density of 1150 kg/m$^3$ (corresponding to a brine with 15 weight % NaCl (Rutishauser et al., 2018)). To compute the hydraulic
head for orthometric heights, the surface and bed elevation were corrected for gradients in the geoid, using the Arctic Gravity
Project geoid (Arctic Gravity Project). Uncertainties of the hydraulic head are derived by propagating the mean crossover
errors in the bed elevation (17.6 m) and an estimated uncertainty of 0.5 m for the ArcticDEM (ArcticDEM), resulting in a
total uncertainty of 4.0 m for the hydraulic head. To identify hydraulically flat areas, we derive the slope of the hydraulic head
($tan \nabla \theta$), which has an uncertainty of 0.5°.



Potential flow paths for subglacial water are derived via application of a flow accumulation algorithm by TopoToolbox (Schwanghart and Kuhn, 2010) to the hydraulic head distribution. In this algorithm flow paths are identified from all grid cells that drain a minimum of 10 upstream cells. Modeled subglacial flow paths are generally very sensitive to variations in the bed topography (e.g. MacKie et al., 2021). To account for uncertainties in our bed DEM, the water routing model is repeated 100 times with randomly perturbed hydraulic heads by adding normally distributed errors with a standard deviation equal to the
hydraulic head uncertainty. Additionally, to test the water routing model for potentially larger uncertainties in the bed DEM towards the ice cap margins where data coverage is reduced, we apply errors equal to two and three times the hydraulic head uncertainty.

A critical implication of a brine-rich subglacial fluid is the increase in relative importance of the bed versus ice surface
topography on the hydraulic gradient as the density of the fluid increases; from ~1/11 (bed/surface topography) for freshwater to ~1/4 for a subglacial brine with a density of 1150 kg/m$^3$ (Rutishauser et al., 2018). Further, we note that the model applied here does not capture spatially varying brine chemistry for example from changes in basal temperature (i.e. cryoconcentration of the brine), evolving drainage morphology through frictional heat dissipation or changes in effective water pressure, or other effects such as thermally constrained (i.e. a cold-ice seal) pathways (Skidmore and Sharp, 1999; Livingstone et al., 2012).

**3 Results**

**3.1 Subglacial bedrock morphology**

While large scale terrain characteristics could be obtained from previous bed DEMs (Dowdeswell et al., 2004; Rutishauser et al., 2018), our dataset and the new, finer resolution DEM reveals additional features in the smaller scale terrain (Figure 3a). This updated DEM shows that bedrock trough T2, which was previously inferred to host a subglacial lake (Rutishauser et al.,
2018), is part of a deeply (~150-300 m) incised canyon that likely extends toward the western margin of the ice cap (Figure 3a) and is overlain by ice exceeding 700 m thickness in large portions of the canyon (Figure 3b). This canyon is one of the most extensive features beneath DIC, and unlike other canyons, does not connect to a marine-terminating outlet glacier under current ice dynamics. In comparison to T2, trough T1 (previously inferred to host the other subglacial lake) is less incised (~100-200 m) into the bedrock and widens towards a plateau area in the north-west, leading into a canyon headward from
Sverdrup Glacier.

The interior part of DIC is underlain by mountainous terrain (hereby referred to as 'the central massif') whereas the southern and western parts of the ice cap (hereby referred as 'lowlands') are characterized by relatively flat bed topography, which is dissected by a few large glacial troughs in the south. The general transition between the central massif and the lowlands can
also be observed in the RMSD-derived basal roughness (Figure 3c), and the bed specularity content (Figure 3d), which is a proxy for the small scale roughness of the glacier bed (Young et al., 2016; Schroeder et al., 2013, 2015).

**Figure 3. Subglacial bedrock morphology beneath DIC**. a) Updated bedrock DEM from this study with 25 m contour lines. Thick black lines mark the boundaries of the glacier catchment areas. b) Ice thickness derived via a subtraction of the bedrock DEM from the ArcticDEM ice surface elevation (ArcticDEM). The location of the previously inferred subglacial lakes in bedrock troughs T1 and T2 (Rutishauser et al., 2018) is marked with blue dashed lines. Background is a Landsat image overlain with the bedrock elevation contours (25 m interval). c) Basal roughness along profile lines expressed as the RMSD. The brown contour marks a RMSD of 25 m. d) Specularity content along the profile lines, overlain with the 25 m RMSD contour line (brown).

## 3.2 Distribution of subglacial water from radar reflectivity

After all corrections are applied, relative basal radar reflectivity values represent a combination of changes in the dielectric contrast at the glacier bed and the roughness of the ice-bed interface (Peters et al., 2005). Although the smooth lowlands show elevated basal reflectivity in general, the highest magnitudes are concentrated in the western part of the central massif (Figure 4a). We interpret variations in the basal reflectivity as transitions between wet and dry bedrock conditions (Chu et al., 2018; Hubbard et al., 2004; Christianson et al., 2012; Peters et al., 2005; Carter et al., 2007; Chu et al., 2016). Based on that, the





observed reflectivity pattern suggests that widespread areas of wet bed lie beneath the central part of DIC. The theoretical contrast in Fresnel reflectivity between wet and dry beds is estimated to be about 10-15 dB (Peters et al., 2005), however, the thresholds that have been used in the literature to differentiate between subglacial water and surrounding dry bedrock are highly variable, ranging between 2 and 26 dB (Wolovick et al., 2013; Carter et al., 2009; Oswald and Gogineni, 2008; Jacobel et al., 2010; Chu et al., 2016). Here, we use a threshold of 12 dB above the mean of all observed reflectivity values in this

study (1.5σ reflectivity anomaly, Figure 4f) to quantitatively identify potential areas of subglacial water (Figure 4b), which is consistent with the theoretical Fresnel reflectivity increase between a dry and wet bed for the dielectric permittivity of seawater (Neal, 1979; Peters et al., 2005). Our results suggest the presence of subglacial water over both areas previously identified as lakes (Rutishauser et al., 2018), however, the reflectivity anomalies over bedrock troughs T1 and T2 extend well beyond the previously outlined lake boundaries, indicating that these may be larger in extent. Furthermore, we observe a prominent cluster

of reflectivity anomalies in the area surrounding T1, which coincides well with the region where salt-bearing rocks are projected to outcrop beneath the ice (Figure 4b).

**3.3 Characterization of water signatures**

The distribution of basal reflectivity suggests the presence of subglacial water over the areas previously inferred as subglacial lakes (Rutishauser et al., 2018), as well as in widespread areas surrounding them (Figure 4b). We evaluate the hydraulic flatness

and specularity content of these water signatures to examine whether they are indicative of 'deep' water bodies or shallow water, including saturated sediments, or dispersed water pockets. While we acknowledge that we cannot specify the depth of water bodies from our dataset, we use the terms 'deep' in a sense of all bedrock undulations over the radar illuminated footprint area (~275 m along-track for our dataset) being submerged by water and the overlying ice column potentially being afloat.

Subglacial water flow and flow direction is generally controlled by the hydraulic gradient, where water can pond in hydraulically flat areas (Shreve, 1972). Thus, high reflectivity anomalies that coincide with hydraulically flat areas are considered typical signatures of subglacial water bodies, a criterion that has been widely used to identify subglacial lakes (Carter et al., 2007; Langley et al., 2011; Bowling et al., 2019; Ilisei et al., 2019; Livingstone et al., 2013). Additionally, as ice-water interfaces are expected to produce specular radar reflections (Schroeder et al., 2015; Carter et al., 2007), observations

of high specularity over hydraulically flat and highly reflective areas are further indicators supporting the presence of a subglacial lake. Here, we define areas of high specularity where values exceed 0.4, corresponding to one standard deviation above the mean of all specularity values over DIC (Table 1).


**Figure 4. Radiometric characteristics and hydraulic slope for derivation of subglacial hydrological conditions.** a) Landsat image (with 25 m bed contour interval) overlain with the basal reflectivity R. Black dotted outlines mark the location of the previously inferred subglacial lakes (Rutishauser et al., 2018). b) Locations where R anomaly exceeds 12 dB, indicating the presence of subglacial water. c) and d) R ≥12 dB, color coded with the hydraulic slope (hslope) and specularity content (Sc). Dots with black outlines show regions with hslope ≤ 0.7° and
Sc ≥ 0.4, respectively. e) Combination of R, hslope and Sc that i) is typical for subglacial lakes (blue), ii) may indicate distributed, shallow water/saturated sediments (orange), and possibly represent areas part of a brine network (black-white dotted outline), and iii) indicate either wet, or dry but highly specular (smooth) bedrock (yellow). Grey shaded area in b-e) represents projected outcrops of the evaporite unit (Ocb). f-g) Distribution of R and Sc over all measured data points, the interpreted brine network and the newly defined lake shorelines.





Our results show that the reflectivity anomalies over bedrock trough T2 are located within a region of low hydraulic slopes (<0.7°) and are characterized by high specularity content (Figure 4c-e). The coinciding high reflectivity, high specularity and low hydraulic slope are consistent with characteristics expected over a subglacial water body, and therefore support the previous inference of a subglacial lake in T2 (Rutishauser et al., 2018). Additionally, the reflecting interface over T2 is exceptionally flat (Fig. S3), which would support the hypothesis of a water filled bedrock trough. We note that although T2

belongs to one of the few areas characterized by such low hydraulic slopes, the hydraulic head over T2 is not perfectly flat (Figs. S4-5). However, this could result from bridging stresses along the valley side walls of this relatively narrow trough, preventing the ice from being fully afloat over subglacial water. Alternatively, the trough could be filled with smooth and potentially water-saturated sediments causing strong and specular reflections. In the absence of a subglacial lake, a relatively flat ice surface causing the observed hydraulic flatness may result from ice flow over bedrock trough T2, where the ice surface

topography is a function of the underlying bedrock perturbations, ice dynamics and ice rheology (Budd, 1970; Raymond and Gudmundsson, 2005). Without being able to fully exclude the above possibilities, the observed signatures over T2 are in good agreement with the physical principles that generally apply over subglacial lakes (e.g. Carter et al., 2007), and we thus argue that the existence of a subglacial lake in T2 is likely. However, the signatures that are indicative of a lake extend well beyond the previously inferred shorelines, from which we interpret new, more extensive shorelines for this subglacial lake (see Section

260 3.4).

In contrast, most of the bedrock trough T1 and its surrounding high reflectivity area are not hydraulically flat, and, except for the T1 trough center, are generally characterized by low specularity (Figure 4c-d). The absence of coincident flat hydraulic heads and high specularity suggests that T1 and its surrounding region consist of shallow water, saturated sediments or

dispersed water pockets rather than "deep" water bodies. The combination of relatively high reflectivity and low specularity in this region could result from a mix of wet (smooth, specular), and dry, rougher interfaces within the radar illuminated footprint. The basal reflectivity can be significantly increased even when the footprint consists of only a small fraction of subglacial water leading to bright, specular interfaces (Haynes et al., 2018). However, where the majority of the footprint is characterized by incoherent surfaces from remaining "small-scale" roughness of non-submerged dry bed, we expect the

specularity content of the returned signal to remain low.

From the combination of these signatures, we interpret that the bedrock in this region is overlain by brine that could be concentrated in small, shallow ponds or channels, saturated sediments, or potentially a brine-slush. Without implying that there is an inherent connectivity between individual components, we hereafter refer to this hydrologic system as a brine network.

The high specularity content within the T1 trough center may result from a locally higher concentration of subglacial water over the radar footprint. Water could be topographically constrained in the center of the trough, causing a larger area of small-scale bedrock roughness to be submerged and therefore appearing locally more specular.



We argue that the spatial correlation between the radar signatures indicating shallow water and the predicted underlying
geology (Rutishauser et al., 2018) further supports our interpretation of a brine network. Brine in this region may be generated
where ice is in direct contact with an underlying salt-bearing evaporite unit (Rutishauser et al., 2018; Harrison et al., 2016;
Mayr, 1980; Thorsteinsson and Mayr, 1987), or from groundwater flow within the salt bearing unit. We approximate an outline
for the hypothesized brine network (Figure 4e) based on the spatial coincidence between the projected salt-bearing rock
outcrops and the high reflectivity (water signature) regions surrounding T1. We note, however, that the outlined area may be
an underestimation of the spatial extent of the brine network, and that the outlined region likely consists of sparsely distributed
wet patches rather than being wet over the entire area.

A few high reflectivity anomalies outside the subglacial lakes and brine network are coincident with high specularity content
(Figure 4d). We cannot fully differentiate between reflectivity anomalies from subglacial water and flat, smooth, or polished
bedrock (Carter et al., 2007). As such, it is possible that these anomalies result from exceptionally smooth and reflective, but
dry bedrock. Similarly, although flat hydraulic heads are a typical criterion for identifying subglacial lakes, hydraulically flat
areas alone are not sufficient evidence for the presence of a subglacial lake. For example, hydraulic flatness can be caused by
a combination of opposing bed and surface slopes, rather than by hydrostatic equilibrium over a subglacial lake. Thus, we
exclude isolated hydraulically flat areas outside of the subglacial lakes in T2 (Figure 4c), of which the majority are
characterized by low specularity, from our interpretation of subglacial lakes.

The high specularity content areas in the western and southern lowlands (Figure 3d) show only slightly elevated reflectivity
(Figure 4a), suggesting that the bedrock in these areas is dry, but exceptionally smooth, and potentially associated with fine
grained sediments or highly polished bedrock (Carter et al., 2007). Such a change in basal characteristics between the massif
and lowland areas could indicate a change in subglacial lithology (Cooper et al., 2019), in which the high specularity content
areas may correspond to outcrops from a specific geological unit (Fig. S6).

**3.4 Geometry of subglacial lakes in T2**

Our dataset suggests that the subglacial lake in bedrock trough T2 is larger than previously inferred (Rutishauser et al., 2018).
We define new lakeshores based on the basal reflectivity and hydraulic flatness (Supplementary Information, Fig. S1), and
identify three water bodies (hereby referred to as Lakes A, B and C) covering a total area of 24.6 km$^2$ (Figure 5). The elevated
mean reflectivity and specularity over these lakes and the significantly reduced standard deviation of basal reflectivity (Table
1, Figure 4b-d) are consistent with signatures of a relatively uniform ice-water interface and therefore support these shorelines.
It is possible that these water bodies are connected in areas not covered by our dataset, or via small channels that remain
undetected in the radar data. We acknowledge that the hydraulic head over these subglacial lakes is not perfectly flat (Fig. S5),





however, it is possible that bridging stresses from the relatively narrow trough prevent the formation of a perfectly flat

hydraulic head.

**Table 1. Radiometric characteristics, subglacial hydraulic slope and areal extent of the subglacial lakes and brine network, compared to the entire dataset.** R is the basal reflectivity normalized to the mean of all measured basal reflectivity values. The values in brackets

represent the standard deviation ($\sigma$) of the corresponding parameter. Additionally, basal ice temperatures were derived from a 1D steady-state advection-diffusion model (Cuffey and Paterson, 2010) with the same parameters as described in Rutishauser et al. (2018) (see Fig. S7 for basal ice temperature map).

|  | Lake A | Lake B | Lake C | Brine network | All data |
|---|---|---|---|---|---|
| Mean R [dB] ($\sigma$) | 11.6 (4.2) | 10.6 (3.8) | 9.7 (3.8) | 7.6 (6.1) | 0 (8.1) |
| Mean specularity ($\sigma$) | 0.5 (0.2) | 0.6 (0.2) | 0.6 (0.2) | 0.1 (0.1) | 0.2 (0.2) |
| Mean hydraulic slope [°] ($\sigma$) | 0.7 (0.4) | 0.7 (0.2) | 0.6 (0.2) | 1.3 (0.5) | 2.2 (2.0) |
| Area [km$^2$] | 11.6 | 4.6 | 8.4 | 169.9 | 5275 (approximate survey grid coverage) |
| Basal ice temperature range (uncertainty estimate) | -15.5°C to -14.4 °C (+/- 4.0 °C) | | | -17.3°C to -15.3°C (+/- 3.5 °C) | - |

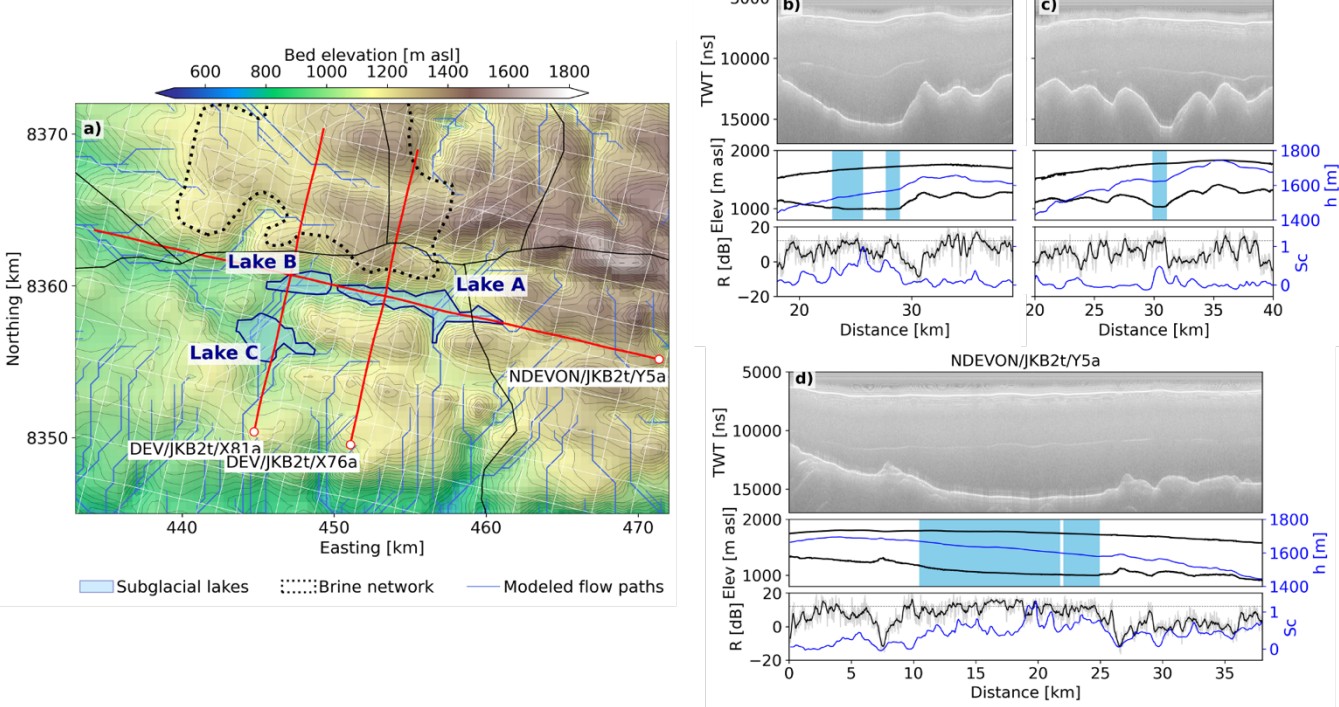

**Figure 5. Geometry of subglacial lakes.** a) Subglacial bedrock topography overlain with the newly defined shorelines of subglacial lakes (blue shaded areas), interpreted outlines of the brine network (black dashed line) and modelled subglacial water flow paths (blue). White



lines represent all SRH1 radar sounding survey profiles, whereas red lines mark the selected profiles shown in b)-d), with the white dots marking the left side of the radargrams. b-d) Example survey profiles over subglacial lakes (blue shaded) showing the radar data (top), the surface and bedrock elevation along with the hydraulic head h (middle) and the reflectivity R and specularity content Sc (bottom).

## 3.5 Modeled subglacial hydrologic pathways

Modeled subglacial flow paths results suggest that if downstream flow occurs, brine from the brine network would likely drain into Sverdrup Glacier, a marine-terminating outlet glacier in the north of DIC (Figure 6). This flow path also persisted across the flow models perturbed with the larger (three times) hydraulic head uncertainty (Fig. S8). In contrast, the subglacial lakes would drain into North- or South Croker Bay Glacier, located in the south of DIC. Here, the flow paths switched from leading into South- to North Croker Bay Glacier when adding higher errors to the model (Fig. S8). Given this uncertainty, it remains unclear to which of these two outlet glaciers brine from the subglacial lakes might drain into. Additionally, the model suggests a possible pathway from the subglacial lakes and brine network towards the land-terminating western margin of the ice cap (Figure 6).

The canyon extending from T2 towards the western margin shows some areas with elevated basal reflectivity (Figure 6a) and specularity content (Figure 6b). It is possible that the signatures result from subglacial water or brine-saturated sediments distributed over this relatively flat and wide subglacial canyon. In contrast, no typical water signatures are observed along the other predicted flow routes downstream of the lakes and brine network. It is unclear whether the bed along these flow paths is indeed dry due to a lack of downstream transport of brine, or whether signatures of subglacial water in these areas simply remain undetected. Brine within the brine-network might be captured in enclosed isolated patches or areas of saturated sediments, limiting its downstream transport. Although no obvious (large-scale) basal freeze-on signatures (Wolovick et al., 2013) are observed in the SRH1 radar data, basal freeze-on may occur over small volumes, which could further limit downstream transport of the brine. However, if downstream transport occurs, it is possible that the brine network transforms from a patchy and distributed water system at the head of the catchment area to a more channelized configuration within the relatively narrow canyons towards the ice cap margins. Small channels may remain undetected in our dataset, depending on the channel size relative to the radar-illuminated footprint, and their orientation to the radar antenna at data acquisition (Schroeder et al., 2015), which could explain the absence of water signatures along the modelled flow paths.



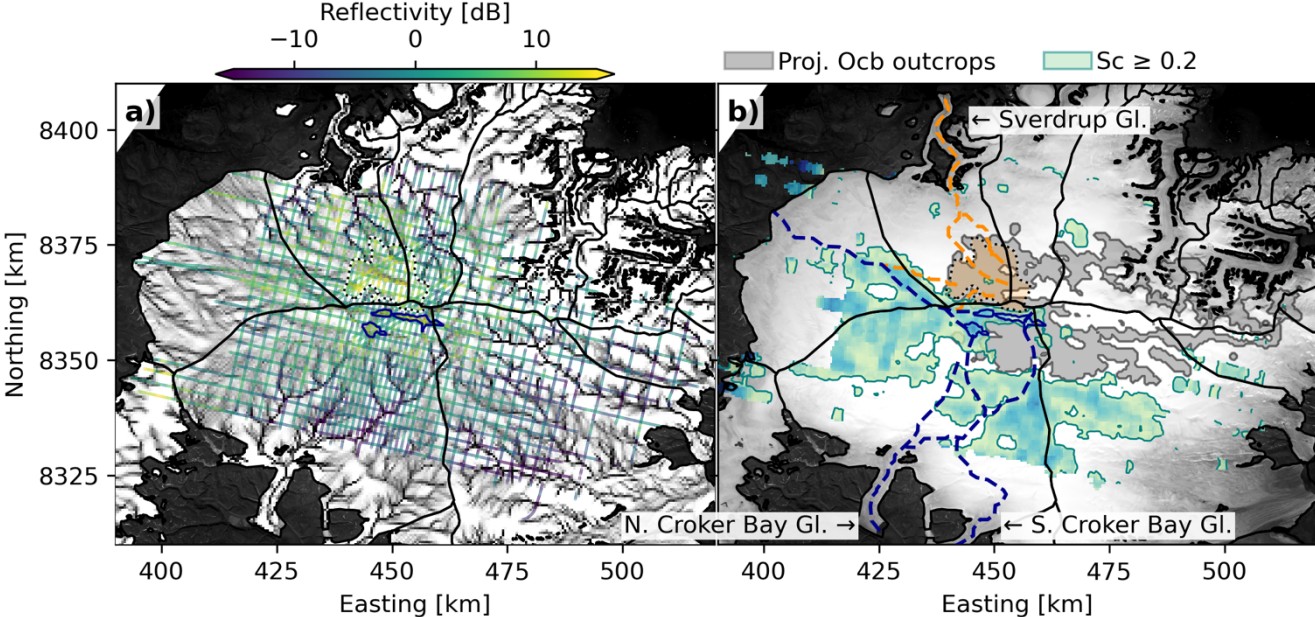

**Figure 6. Hydrological pathways.** a) Modeled potential subglacial hydrological pathways (grayscale) beneath DIC, overlain with the basal
reflectivity. b) Landsat image overlain with areas where the Ocb unit is projected to outcrop beneath the ice (gray), areas of elevated
specularity content Sc (increasing values with darker color), and the main water routes where brine from the subglacial lakes (blue) and brine
network (orange) potentially propagates downstream.

## 4 Discussion

Our results support the previous evidence for a subglacial lake in bedrock trough T2 (Rutishauser et al., 2018) and indicate
that this feature likely consists of three subglacial lakes, with a larger areal extent (total of 24.6 km$^2$) than previously estimated.
The length of subglacial lake A, the longest of the lakes is about 11 km, which is just above the typical length of most subglacial
lakes identified beneath Antarctica (<10 km length) (Wright and Siegert, 2013) and Greenland (<0.5 – 6 km length) (Bowling
et al., 2019).

We also find evidence consistent with an extensive brine network consisting of shallow water or saturated sediments, including
the area previously inferred as a subglacial lake in bedrock trough T1. The possibility of shallow water in T1 was acknowledged
by Rutishauser et al. (2018) but could not be resolved with certainty due to the relatively sparse data coverage available. This
highlights the importance of collecting closely spaced geophysical datasets over relatively small features such as the Devon
subglacial lakes. The combination of high relative reflectivity and low specularity over the brine network resembles signatures
that have previously been referred to as "fuzzy lakes", and have been interpreted as saturated sediments and possibly small
ponds of subglacial water (Carter et al., 2007).



Although the presence of subglacial water is generally thought to play a crucial role in ice dynamics (Siegert and Bamber, 2000; Fricker and Scambos, 2009; Stearns et al., 2008), the brine network and subglacial lakes are located where ice flow is
predicted to occur from internal deformation alone (Van Wychen et al., 2017; Burgess et al., 2005; Paterson and Clarke, 1978). We argue that the generally slow ice flow velocities over the brine network region provides additional evidence that the brine network consists of small patches of wet bed surrounded by areas where ice is frozen to the bed and thus generates enough friction to prevent basal sliding. Similarly widespread areas of wet beds, including saturated sediments or shallow ponded water, have been inferred beneath slow-moving interior areas of the Antarctic (Carter et al., 2009; Zirizzotti et al., 2012, 2010;
Laird et al., 2010) and the Greenland (Oswald and Gogineni, 2008, 2012; Oswald et al., 2018; Jordan et al., 2018) ice sheets. While we can derive the general direction of potential subglacial flow paths via a simple water routing model, detailed configuration of the hydrologic system and brine network beneath DIC (i.e. isolated vs. connected patches of brine/saturated sediments, distributed vs. channelized system) remain largely unknown and are subject to further investigation.

Further, our study shows a spatial correlation between the independently derived subglacial water signatures of the brine network and projected outcrops of a potentially salt-bearing evaporite unit (Rutishauser et al., 2018; Harrison et al., 2016; Mayr, 1980; Thorsteinsson and Mayr, 1987), supporting the previous hypothesis for the source of hypersaline subglacial fluids. We speculate that the characteristics of the brine network are related to, and potentially controlled by the bedrock lithology, where the sub-ice outcrops of salt-bearing evaporite rocks previously proposed (Rutishauser et al., 2018) play a crucial role in
the formation of the hypersaline fluid and its geochemistry.

A recent study highlights the importance of considering the electrical conductivity of subglacial materials when interpreting basal radar reflectivity (Tulaczyk and Foley, 2020). In particular, Tulaczyk and Foley (2020) demonstrate that high-conductivity material such as clay-bearing sediments can cause reflectivity anomalies as high as subglacial water surrounded
by dry bedrock, and thus could lead to misinterpretations regarding the presence of subglacial water and lakes from radar sounding data. Due to the spatial association between high basal reflectivity over the brine-network and projected outcrops of the evaporite unit (Rutishauser et al., 2018), we consider the existence of brine-saturated sediments/shallow brine as the most plausible explanation for the observed radar reflectivity patterns. Furthermore, the combination of observed characteristics over T2 (high reflectivity, high specularity, hydraulic flatness) are in good agreement with the physical principles that generally
apply over subglacial lakes (e.g. Carter et al., 2007). However, the possibility that the high reflectivity anomalies beneath DIC may arise from saturated (clay-rich) sediments (Tulaczyk and Foley, 2020) or from highly polished, exceptionally flat and smooth but dry bedrock (Carter et al., 2007) cannot be neglected outright. Future investigations using other geophysical techniques such as seismics (Peters et al., 2008; e.g., Horgan et al., 2012), transient electromagnetics or magnetotellurics (e.g. Hill, 2020; Key and Siegfried, 2017; Killingbeck et al., 2020; Mikucki et al., 2015) techniques could resolve the remaining
uncertainty about the existence and distribution of hypersaline subglacial water beneath DIC.



One subglacial hydrological system with comparable hypersaline conditions lies beneath Taylor Glacier, Antarctica (Lyons et al., 2005; Mikucki and Priscu, 2007; Badgeley et al., 2017; Hubbard et al., 2004; Lyons et al., 2019; Mikucki et al., 2004, 2015). Here, subglacial brine has been observed to remain liquid at basal ice temperatures of -17 °C through a combination of
freezing point depression from the hypersaline conditions, and partial freeze-on of brine which results in warming of the surrounding ice through latent heat release and a further increase in brine-salinity through cryoconcentration (Badgeley et al., 2017). Although basal processes and interactions between the brine, underlying rocks, and the overlying ice remain unclear, it is possible that cryoconcentration and latent heat release processes upon basal freeze-on contribute to sustaining the brine beneath DIC liquid at basal ice temperatures as low as -17.5 °C (Table 1).


The hypersaline subglacial discharge at Taylor Glacier, Antarctica has been shown to contain viable microbes (e.g. Mikucki et al., 2004) and thus areas of the glacier bed covered by such fluids are considered as microbial habitat. By inference, the area of the bed beneath the DIC that is potentially covered by a ~ 170 km$^2$ brine network, is substantive and expands the potential subglacial microbial habitat beneath DIC. However, it is noted that brine at the bed only covers a portion of this area, and that
the brine network would likely comprise a heterogeneous mixture of environments/habitats. These could include (i) brine pockets of a range of sizes, but generally of shallow nature, that could contain partial sedimentary fill, and (ii) saturated sediments of varying (but unknown) thickness. Depending on brine availability and the configuration of the subglacial hydrological system, there may be a degree of interconnectivity between individual components within this brine network following the hydraulic gradients. This contrasts with the proposed subglacial lake system comprised of a few larger volume
water-body components.

The nature and connectivity of a subglacial hydrological system has been identified as a key variable in determining geochemical weathering, and the redox potential of specific environmental niches in freshwater subglacial systems, and this impacts the range of metabolic capabilities of microorganisms that can inhabit those niches (Tranter et al., 2005). This would
also be the case for a hypersaline system, but microbes in any of the subglacial environments of DIC would need to be adapted to high salinity and low temperatures. Potential differences in the underlying lithology between regions identified as likely hosting subglacial lakes (Eleanor River Formation, Oe) and a brine network (Bay Fiord Formation, Ocb) could also impact chemolithotrophic energy sources (Fig. S6). Both of these formations have limestone and dolostone components that could provide organic material to the subglacial systems, but the Eleanor River Formation has been documented to contain pyrite in
outcrops to the west of the DIC (Mayr et al., 1998). Pyrite has been demonstrated as an important energy source for microbes in subglacial environments (e.g. Mitchell et al., 2013; Montross et al., 2013) and its presence (or absence) has the potential to influence aspects of subglacial microbial community composition (Skidmore et al., 2005). Collectively the complex mixture of physical environments and bedrock lithologies likely results in a diverse range of subglacial microbial habitat beneath DIC.



The Devon subglacial lake complex has already been identified as a terrestrial analog for potential brine habitats inferred on other planetary bodies (Rutishauser et al., 2018). The diverse subglacial hydrological environments beneath DIC proposed here represent analogs for a spectrum of sub-surface briny bodies on other icy worlds. Features observed at the surfaces of icy ocean worlds are consistent with the presence of near-surface fluid bodies (Waite et al., 2009; Schmidt et al., 2011; Postberg et al., 2011; Michaut and Manga, 2014; Walker and Schmidt, 2015; Manga and Michaut, 2017; Steinbrügge et al., 2020), that
represent conditions potentially habitable for microbial life, and are thus high-valued targets for future exploration.

On Europa, the formation of chaos terrain has been proposed to be a direct consequence of the evolution of such near-surface saline fluid bodies, and may also generate brine networks within the neighboring ice regolith (Schmidt et al., 2011). Thus, conditions hypothesized in fluid systems within the ice shell of icy moons, including perched lakes, may be analog to, and
could therefore be constrained via the multifaceted hypersaline subglacial hydrological environment at DIC. The identification of perched lakes on other icy worlds is of particular interest as they could be associated with cryovolcanic activity (Postberg et al., 2011; Sparks et al., 2017; Jia et al., 2018; Steinbrügge et al., 2020), therefore representing locations with an increased probability for plume material which could be sampled by an orbiting spacecraft. Additionally, potential outflows of brine and associated microbial communities into the ocean beneath marine terminating outlet glaciers of DIC would represent an analog
environment for briny fluids that may drain from near-surface perched lakes of Europa's ice shell to the underlying global ocean (Hesse et al., 2020).

The subglacial lakes and brine network could also represent an analog for a layer of brine-slush formed in the final stages of the freezing of a sub-ice ocean (Zolotov, 2007) or hypersaline fluids beneath the southern polar ice cap on Mars (Orosei et al.,
2018; Lauro et al., 2020). Like the hypersaline water system beneath DIC, the stability of water beneath Mars' southern polar ice cap has been attributed in part to freezing point depression by salts sourced from underlying rock (Orosei et al., 2018; Arnold et al., 2019; Lauro et al., 2020). Although liquid water is not stable on the surface of Mars today, shallow brine networks are thought to be a widespread and significant potential microbial habitat (Jones, 2018). Impact sites on Europa (Steinbrügge et al., 2020) and Mars (Michalski et al., 2013; Martín-Torres et al., 2015) have been identified as locations where transient
hydrological systems could form.

## 5 Conclusions

The study presents results from a targeted aerogeophysical survey over previously hypothesized subglacial lakes located in two bedrock troughs (T1 and T2) beneath DIC (Rutishauser et al., 2018). We use a combination of radar derived basal reflectivity and specularity content, and the hydraulic flatness to evaluate the initial hypothesis of the two subglacial lakes,
examine their full extents and characterize the surrounding subglacial hydrological environment. Our results support the previous evidence for one of the subglacial lakes (located in bedrock trough T2) and suggest that this feature consists of three

distinct water bodies with a total areal extent of 24.6 km², which is larger than previously estimated. On the contrary, we conclude that the area over bedrock trough T1 previously outlined as a subglacial lake likely consist of shallow water. This possibility was acknowledged by Rutishauser et al. (2018) but could not be resolved with certainty due to the relatively sparse

data coverage. Lastly, we find evidence consistent with an extensive brine network covering a total area of ~170 km², where brine maybe concentrated in small, shallow ponds/channels or saturated sediments.

Overall, our results reveal that the subglacial hydrological conditions beneath DIC are more complex than previously suggested. Although the formation and detailed configuration of the subglacial lakes and brine network remain unknown, their

cold and hypersaline conditions could facilitate microbial habitats that are likely analogous to briny habitats on other planetary bodies. Furthermore, as remote characterization of the subglacial hydrology on other icy worlds is an important initial step towards in situ sampling by a spacecraft, lander, or submersible platform, the subglacial hydrological system beneath DIC represents an analog environment for technology development towards the exploration of similar potential habitats on other icy worlds. Results from this study will help inform the planning of future investigations of this potentially unique subglacial

hydrological environment, including in-situ access and sampling of the subglacial brine to explore its habitability for microbial life.

**Data availability**

We are currently preparing the data supporting the findings from this study for publication via zenodo.org. Datasets will include radargrams, ice surface elevation, bedrock elevation, ice thickness, basal reflectivity and specularity for each profile.

Additionally, we will submit georeferenced tif files for the ice thickness, the bed DEM, subglacial hydraulic head, and modeled hydrological flow paths, as well as shapefiles for the lakes and brine network.

**Author contribution**

A.R. led the geophysical data analysis and wrote the manuscript. A.R., D.D.B., D.A.Y, L.H.B., N.S.W., A.D. and A.S.C. planned the study and contributed to data collection. D.A.Y processed the geophysical data and derived the specularity content.

M.L.S provided guidance on the geological interpretations and assessment of microbial habitat potential, and N.S.W. provided guidance for the planetary analog assessment. All authors contributed to the data analysis and interpretation of results and provided inputs for the manuscript.

**Competing interests**

The authors declare that they have no conflict of interest.




## Acknowledgments

The aerogeophysical survey and subsequent standard data processing were funded by the Weston Family Foundation. We also thank the G. Unger Vetlesen Foundation and the UTIG Postdoctoral Fellowship program who provided further funding for data analysis. M.L.S. was partially supported by NASA NNX16AJ64G and NASA 80NSSC20K1134. We thank PCSP and

Kenn Borek Air Ltd. for logistical support, and the Nunavut Research Institute and the peoples of Grise Fjord and Resolute Bay for permission to conduct airborne surveys over Devon Ice Cap. Finally, we thank Dillon Buhl, Gregory Ng and Tom Richter for help with data collection and processing, Scott Kempf for assistance with data processing, and Sam Christian and Miguel Liu-Schiaffini for help with radar reflection picking. This is a UTIG contribution.

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
