# Peer review of "Radar sounding survey over Devon Ice Cap indicates the potential for a diverse hypersaline subglacial hydrological environment"

_The Cryosphere, 2021_

## Author Comment (AC1)

**Author Responses** for manuscript "Radar sounding survey over Devon Ice Cap indicates the potential for a diverse hypersaline subglacial hydrological environment"

Referee comment #1

*This is an interesting and well-written paper, and I believe it will make a great addition to this journal. I have identified a few minor issues that I have noted below, but pending these, I believe that the article is suitable for publication.*

We thank the referee for the great and constructive feedback which helped to improve the manuscript.

**Specific comments:**

*Line 163: why only 100 times? Usually several thousand times are chosen*

We agree that we should test the subglacial hydrology model with more model runs. We therefore changed the number of model runs to 1000. The overall result did not change significantly but helped to identify the main flow paths a bit more clearly. In particular, while a few model runs still predict water flow towards South Croker Bay Glacier, we think that the uncertainty in the flow path at this location is reduced when using the 1000 model runs. Accordingly, we slightly edited the text on L329-331:

"In contrast, the subglacial lakes would most likely drain into North Croker Bay Glacier, located in the south of DIC. However, in a few of the model runs, the flow paths switched from leading into North- to South Croker Bay Glacier (Fig. S12). Given this uncertainty, we cannot conclusively determine to which of these two outlet glaciers brine from the subglacial lakes might drain into."

*Figure 3: Is there a reason that the 25m RMSD contour line is highlighted? If so, state in the caption or main text.*

The 25 m RMSD was chosen based on a visual correlation with specularity anomalies. We added a statement in brackets to the figure to elaborate:

"c) Basal roughness along profile lines expressed as the RMSD. The brown contour marks a RMSD of 25 m (chosen based on visual correlation with specularity anomalies)"

*Line 290: Could these anomalies not be further pockets of brine?*

Yes, we realized that this possibility wasn't noted explicitly, and therefore changed the sentence to:

"These could represent additional areas with brine or brine-saturated sediments, however, we cannot fully differentiate between reflectivity anomalies from subglacial water and flat, smooth, or polished bedrock (Carter et al., 2007)."

*Figure 4: How was the extent of the hypothesized brine network chosen? There are many isolated anomalies (orange in Fig. 4e) within the location of the Ocb outcrop that are not identified as brine networks. How/why was this discrimination made?*

The outline of the brine network was chosen to include the areas with the highest concentration of reflectivity anomalies over the Ocb unit. To clarify, we added the following sentence:

"We limit the outline of the brine network to the T1 region where the highest concentration of reflectivity anomalies coincides with the Ocb unit, however, we note that there are isolated reflectivity anomalies outside this area which also coincide with the Ocb unit."

*Table 1 caption. 'brackets' should be 'parantheses'*

Done

---

## Author Comment (AC2)

**Author Responses** for manuscript "Radar sounding survey over Devon Ice Cap indicates the potential for a diverse hypersaline subglacial hydrological environment"

Referee comment #2: Slawek Tulaczyk

*This manuscript reports on the results of a regional radar survey of the Devon ice cap. The data are interpreted in the context of prior work by the same team in which they proposed the presence of briny subglacial lakes in this study region. The results presented here broadly corroborate this prior contention. However, the authors also see evidence for extensive areas that are not lakes but likely brine-saturated sediments. Generally, the new dataset yields a detailed picture of the complex subglacial hydrology beneath the Devon ice cap and may be helpful in a range of glaciological and microbiological studies in the future. The manuscript includes an excellent discussion of the possible microbial habitats beneath the Devon ice cap and overviews their relevance to search for extraterrestrial life on icy planetary bodies.*

*Overall, I find this manuscript to be informative and worth publishing. I have several specific comments on the methods and results sections (see below). However, once I got to the discussion and results sections, they seemed to be very well crafted with proper caveats stated where they are necessary. The figures are loaded with information and clear. Citations are extensive and relevant. I did suggest a few additional papers that the authors may want to take into account. I do think that there are several places where more methodological rigor can be introduced and a few places where statements can be made in a more even-handed way.*

We thank Dr. Tulaczyk for the detailed and constructive feedback to help improve this manuscript. We especially appreciate the comments about the conductivity effects on basal reflectivity, which we hope we have now addressed through more consistent elaboration throughout the manuscript.

**Specific comments:**

*Line 57: It is much more relevant to this study to give the radar wavelength in ice, not in air.*

We agree and changed the text to "(2.8 m wavelength in ice)"

*Line 83: The significant difference between the median and the mean suggests that the cross-over errors are not normally distributed. I would be interested in including a histogram of the errors as a figure. Alternatively, the authors should at least point out the deviation of this distribution from the Gaussian distribution. This is important because it is enough for a standard normal distribution to give the mean and the standard deviation to describe the underlying probability distribution function fully. However, more complex distributions (e.g., multimodal, asymmetric, or skewed distributions) need to be described by more than just the two first moments. With non-Gaussian distributions, one also cannot assume that 68% and 95% of data fall within one and two standard deviations from the mean.*

We agree that this was not very clear. We used the absolute values of the crossover errors, that is why the distribution was not a normal distribution. We added a figure to the Supplementary

Information with the histogram and report the standard deviation of the distribution as uncertainty. We changed the sentence to:

"A total of 47233 crossover points reveal a mean crossover error of 0.1 m in nadir ice thickness (mean absolute error of 17.6 m) with a standard deviation of 46.0 m (Fig. S5)"

[Figure]

**Figure S5. Distribution of bed elevation errors at radar sounding crossover points.** A total of 47233 crossover points reveal a mean error of 0.1 m with a standard deviation of 46 m.

*Line 94: This statement is not incorrect, but, in the context of this study, it sets up an unrealistic juxtaposition of subglacial conditions. Why would the subglacial bedrock be dry in the study area if there is so much water in lakes and sediments in the region? Ice over bedrock saturated with brine could produce higher radar reflection strength than ice over a subglacial lake filled with very fresh meltwater (e.g., Tulaczyk and Foley, 2020).*

We agree that it's important to mention the effect of electrical conductivity here. We therefore added the sentence "However, Tulaczyk and Foley (2020) demonstrate that subglacial material with high electrical conductivity (e.g. clay, brine-saturated sediments) can produce equally strong reflections as subglacial lakes filled with fresh water. Thus, it is important to consider both changes in water content and electrical conductivity when interpreting basal reflectivity"

*Line 113: It is entirely unclear how the authors choose where to draw the line designating Nh. Why is it chosen to be at such a value of correlation coefficient (ca. 0.1)? Why not at 0.2 or 0.3? Is the correlation coefficient magnitude here R or R-squared. Estimation of the attenuation coefficient is an important methodological step, and it needs to be fleshed out more than it is here.*

We agree that the attenuation rate uncertainty is likely higher than estimated by using a correlation coefficient of 0.1 following (Schroeder et al., 2016). We now report the correlation coefficient (r) of the linear fit and calculate the root-mean-square error (RMSE) between the regression line and

the observed bed power values. We change the paragraph to the following and updated the figure to include the linear regression statistics.

"We constrain our fit to ice thicknesses between 50-650 m where the relationship appears most linear, resulting in a one-way attenuation rate (slop of the fit) of 21.8 dB/km, with a correlation coefficient (R) of -0.57 (Figure 2). The root-mean-square error (RMSE) between the regression line and the observed bed power is 8.0 dB, indicating relatively large attenuation rate uncertainties. This is likely due to processes affecting the radar attenuation rate that are not accounted for in this simple regression method, including the presence of subglacial water (increases the bed power for a given ice thickness), heterogeneous distribution of ice temperature and chemistry, as well as the ice surface and bedrock roughness (decreases the bed power for a given ice thickness). Here, we do not attempt to further constrain the variability in attenuation rates and use the 21.8 dB/km attenuation rate to correct the observed bed power over the entire dataset."

[Figure]

**Figure 2. Derivation of englacial attenuation rates**. Correlation and linear fit (red) between the ice thickness and the geometrically corrected bed power of the SRH1 dataset over DIC, along with contour lines indicating the density distribution. The attenuation rate is derived from the slope of the fit (21.8 dB/km).

*Figure 2: It is clear from the distribution shown in this figure that quite a range of quite different lines, representing different attenuation coefficients, could be fit reasonably well into this distribution. For one, there is a distinct peak to this distribution (at ca. -42dB and 480m), with the resulting linear fit being determined by extreme tales at low and high frequencies. I'm sorry, but I do not buy the relatively narrow uncertainty of 3.1 dB/km that the authors assign to their fit. Linear regression can be performed in a way that yields the uncertainties in the two parameters that define the linear fit (one of which is the slope of the line, from which attenuation is calculated). The authors need to report these statistically-based uncertainties on their attenuation estimates.*

*The validity of the method used by Schroeder et al. (2016) may be dependent on the specific distribution of the bed power vs. thickness. In this case, this distribution does not support a tight linear fit.*

Agreed, see changes made in response to the previous comment.

*Lines 131-132: As in the dry bedrock example, the authors use an extreme end-member here and create a false sense of the singular relationship between high specularity and ice- on-water interfaces. In reality, ice overriding sediments can easily have a flat interface, particularly if the sediments are weak. However, I have seen even deglaciated granitic surfaces that are smooth enough on the scale of 1-10m that they would have high specularity.*

To clarify that smooth, non-water interfaces could generate high specularity as well, we added the following sentence: "However, we note that weak sediments or highly polished bedrock that has a flat and smooth interface on a wavelength scale could produce a similarly high specularity as an ice-water interface."

*Line 156: The mean cross-over error of 17.6 m itself is associated with uncertainty. In reality, you do not know if it is 17.6 m or some larger or smaller value. This is why the concept of uncertainty propagation has been invented. Please, calculate the uncertainty of the mean cross-over error and propagate it into your calculation of uncertainties of the hydraulic head.*

Agreed. We now incorporate the standard deviation of the bed elevation crossover errors and propagate it into the hydraulic head/slope uncertainty.

"Uncertainties of the hydraulic head are derived by propagating the standard deviation of the crossover errors in the bed elevation (46.0 m) and an estimated uncertainty of 3.7 m for the ArcticDEM (Supplementary Information), resulting in a total uncertainty of 12.3 m for the hydraulic head."

This, together with the higher uncertainty estimate for the ArcticDEM results in a much more conservative error estimate and significantly increases the hydraulic head/slope uncertainty (see below).

*Line 157: Similarly, the uncertainty of 0.5 m for the ArcticDEM seems unrealistic from my experience. The authors should look at independent estimates rather than just quote a potentially optimistic estimate produced by those who generated ArcticDEM. For instance, the analysis of Xing et al. (2020) suggests a 2-sigma error of about +-5 meters for the Greenland ArcticDEM analyzed by them.*

We agree that the 0.5 m estimate is likely an optimistic estimate. To evaluate the ArcticDEM ice surface, we use the laser altimetry data collected during the same aerogeophysical campaign (SRH1). The comparison revealed a mean offset of 0.7 m, and a standard deviation of 3.7 m. We replaced the 0.5 m uncertainty estimate with 3.7 m (std) and propagate this error to the hydraulic head and slope uncertainty. We added the following section to the Supplementary Information:

"To evaluate the ArcticDEM ice surface (100 m composite) as input for the calculation of the subglacial hydraulic head/slope, and to estimate the uncertainty of the ArcticDEM over DIC, we compare the ArcticDEM to laser altimetry measurements (with a rms crossover error of 21 cm) conducted during the SRH1 aerogeophysical survey. Our analysis reveals a mean offset of 0.7 m, with a standard deviation of 3.7 m (Fig. S2). We use the 3.7 m standard deviation as an uncertainty estimate for the ArcticDEM-derived ice surface elevation, which is above the 0.5 m estimate reported by the ArcticDEM providers (ArcticDEM)."

[Figure]

**Figure S2. Comparison of ArcticDEM to laser altimetry measurements.** The distribution of offsets between airborne laser-altimetry ice surface elevation measurements and the ArcticDEM over Devon Ice Cap reveals a mean offset of 0.7 m with a standard deviation of 3.7 m.

*Lines 200-201: This statement is somewhat misleading. The paper by Peters et al. (2005) uses complex permittivity, which incorporates the effects of real permittivity and electrical conductivity on radar reflectivity of the bed. Yet, later (Line 203), the authors interpret the observed spatial variations in the basal reflectivity as transitions between wet and dry bedrock conditions. Whereas this simplified interpretation is commonly used, it is not strictly correct, as pointed out in Tulaczyk and Foley (2020) and in the open discussion to this TC paper. Spatial variations in relative bed reflectivity may be due to changes in the salinity of subglacial water, which influences the imaginary (electrical conductivity) component of complex permittivity. The relative amount of subglacial water (i.e., the wet vs dry language used here), influences the real part of complex permittivity. In reality, the authors could be looking at subglacial materials with the same amount of water but seeing very different radar reflectivities because in some places, this water is fresh meltwater, and in others, it is hypersaline brine. The effect of electrical conductivity seems particularly pertinent to this study, given that the authors propose that very hypersaline brines (five times more saline than seawater) are present in some parts of their study region. The role of*

*electrical conductivity in the radar reflectivity of glacier beds needs to be tackled in this manuscript head-on. The authors cannot say in some parts of this manuscript that there are very saline (conductive) brines in the region and then default to interpreting bed reflectivity as if only the real part of complex permittivity mattered in radar reflection.*

We agree that the manuscript should tackle the conductivity effect on radar reflectivity earlier on than simply mentioning it in the discussion section. We changed the paragraph (L201-205) to the following:

"After all corrections are applied, relative basal radar reflectivity values represent a combination of changes in the dielectric permittivity (i.e. presence of water) and electrical conductivity (i.e. salinity, presence of clay) at the glacier bed, as well as the roughness of the ice-bed interface (Peters et al., 2005; Tulaczyk and Foley, 2020). Although the smooth lowlands show elevated basal reflectivity in general, the highest magnitudes are concentrated in the western part of the central massif (Fig. 4a). Excluding the effects of electrical conductivity of a subglacial material (Tulaczyk and Foley, 2020), variations in basal reflectivity have typically been interpreted as transitions between wet and dry bedrock conditions (Chu et al., 2018; Hubbard et al., 2004; Christianson et al., 2012; Peters et al., 2005; Carter et al., 2007; Chu et al., 2016). Based on that, the observed reflectivity pattern suggests that widespread areas of wet bed lie beneath the central part of DIC. However, the observed variations in relative bed reflectivity could also result from changes in electrical conductivity of the subglacial material (rather than dry-wet transitions), including changes in the salinity (e.g. freshwater versus brine) of subglacial water or saturated sediments (Tulaczyk and Foley, 2020). While some variations in brine salinity likely occur, the existence of freshwater over the study area is unlikely due to estimated basal ice temperatures well below the pressure melting point (Fig. S7, (Burgess et al., 2005; Van Wychen et al., 2017; Paterson and Clarke, 1978)). Furthermore, we cannot rule out that high basal reflectivity anomalies are caused by dry, but highly conductive bed material (further detailed in the Discussion section). Nevertheless, we hereafter interpret high relative reflectivity anomalies as areas consisting of saline subglacial water (brine), and in the form of saturated sediments."

*Line 221: There is literature focused on estimating lake bathymetry from their size and the topography of lake surroundings (e.g., Heathcote et al., 2015). Perhaps the authors can lean on such prior research to get a better handle on lake depths?*

We agree that estimating lake bathymetry from the valley walls could be done, but we believe this is beyond the scope of this manuscript.

*Line 231: Can you quantify how 'flat' or 'rough' on the scale of the radar wavelength does the ice bed have to be to yield specularity of >0.4? Just from intuition, it seems that 0.4 can still correspond to pretty rough surfaces. I am not that comfortable associating specularity of such low magnitude with only being consistent with ice-water interfaces. Again, ice-on-weak-sediments can make pretty flat interfaces too. Moreover, such an interface can be highly reflective. Particularly when the subglacial fluid is hypersaline and/or if the subglacial material contains clays (both are*

*electrically conductive materials). Furthermore, an extensive area of weak subglacial sediments would lead to the ice surface slope being shallow, which would also result in hydraulic flatness. All three criteria for subglacial lakes used here can also indicate wet, weak sediments.*

It is difficult to quantify a basal roughness from the specularity content directly. However, to demonstrate why we believe that the 0.4 threshold is valid for the identification of smooth (and potentially water covered) areas, we added the following section and figures to the supplementary material.

**"3 Specularity content threshold justification**

We use a specularity content threshold of 0.4 for declaring a detection of possible subglacial water. Theory suggests that value for an individual detection should be as high as 0.9 (Schroeder et al., 2015). However, we allow for a lower threshold in the context of the filtering process necessary when deriving the specularity content (Fig. S3b) and the noise in the bed power difference between the 1D and 2D focused products (Fig. S3c), to which the specularity content algorithm has a non-linear response (Fig. S4). Noise between the 2D and 1D bed power difference can be caused by several sources, including instability in the semi-automatic bed picking, differences in the horizontal resolution between the two focusing aperture lengths, and clutter. Our results show that even over specular areas, such as the area over the hypothesised subglacial lake in T2, considerable small-scale variations (noise) in the 1D and 2D focussed bed power occur (Fig. S3c). The noise is then propagated into the specularity algorithm (Schroeder et al., 2015, Eq. 6), where the relationship between the specularity content and bed power difference becomes hyperbolic where the 2D focused bed power is smaller than the 1D focused bed power (Fig. S4). Using the standard deviation of the 2D-1D focused bed power over the hypothesized lake area as an uncertainty estimate (grey shaded area in Fig. S3c), it becomes clear that the 0.4 specularity threshold is well above the lowest expected specularity value (0.13) within this uncertainty band (Fig. S4).

[Figure]

**Fig S3. Specularity content along a sample profile (IRSPC2_2018153_NDEVON_JKB2t_Y5a.txt).** a) Radargram of a transect flown along the bedrock trough T2 and the hypothesized subglacial lake. b) Raw (blue dots) and filtered (black line) specularity content. The blue shaded rectangle marks the area identified as subglacial lakes (A and B) in this study. The purple shaded box shows where we count the filtered data towards possible water detection (specularity content above 0.4).

[Figure]

**Figure S4. Relationship between the specularity content and the difference in bed power from the 2D and 1D focused products.** The specularity content is derived following Eq.6. in Schroeder et al. (2015) using 1D and 2D aperture lengths of 700 m and 2011 m, respectively, an aircraft height above ice of 500 m, and an ice thickness of 800 m. The grey shaded area shows the uncertainty range in 2D minus 1D focused bed power derived over the subglacial lake candidate (Fig. S3c). The purple shaded area shows the specularity content values above the 0.4 threshold, potentially indicating areas of subglacial water.

In regards to an alternative explanation for the high specularity content over T1, we address the possibility that T2 consist of smooth bedrock/sediments rather than a subglacial lake on L252: "Alternatively, the trough could be filled with smooth and potentially water-saturated sediments causing strong and specular reflections. In the absence of a subglacial lake, a relatively flat ice surface causing the observed hydraulic flatness may result from ice flow over bedrock trough T2, where the ice surface topography is a function of the underlying bedrock perturbations, ice dynamics and ice rheology (Budd, 1970; Raymond and Gudmundsson, 2005)".

*Line 286: Can you elaborate why you do not think that there is a widespread area of wet conditions here?*

We believe that the combination of high reflectivity and low specularity indicate that there is some water over the radar footprint (causing high reflectivity), but not covering the entire radar footprint (leaving rough patches causing the low specularity). We introduce this concept on L266. To clarify, we added:

"We note, however, that the outlined area may be an underestimation of the spatial extent of the brine network, and that the outlined region likely consists of sparsely distributed wet patches rather than being wet over the entire area (interpreted from the combination of high reflectivity and low specularity content)."

*Lines 297-298: What about the possibility that less saline fluids (e.g., fresh subglacial meltwaters) cause the low bed reflectivity in this high specularity area? This seems like an equally plausible explanation as compared to the 'dry bedrock' one.*

To address this possibility, we included the following section:

"Alternatively, these areas could be covered by fresh subglacial water causing a specular interface, whereas the reflectivity increase towards the center of the ice cap can be explained by changes from freshwater to brine (Tulaczyk and Foley, 2020). However, given that the estimated basal ice temperatures (Fig. S7) are well below the pressure melting point of ice (Burgess et al., 2005; Van Wychen et al., 2017; Paterson and Clarke, 1978), and that the ice flow is slow (see further elaboration in the Discussion Section), it is unlikely that such large areas beneath DIC are covered by subglacial freshwater."

*Lines 454-455: Three papers have been published in GRL this summer, casting doubt on the interpretation of high reflectivity anomalies beneath the Martian SPLD as liquid brines (Bierson et al., 2021; Khuller and Plaut, 2021; Smith et al., 2021*

We agree that we should include these new studies. We therefore changed the section to:

"The subglacial lakes and brine network could also represent an analog for a layer of brine-slush formed in the final stages of the freezing of a sub-ice ocean (Zolotov, 2007) or hypersaline fluids beneath the southern polar layered deposits (SPLD) on Mars (Orosei et al., 2018; Lauro et al., 2020), although the existence of liquid brines beneath the Martian SPLD has been challenged by alternative interpretations that do not include the presence of liquid brine (Bierson et al., 2021; Khuller and Plaut, 2021; Schroeder and Steinbrügge, 2021; Smith et al., 2021)"